# Triboelectric Nanogenerators for Preventive Health Monitoring

**DOI:** 10.3390/nano14040336

**Published:** 2024-02-08

**Authors:** Mang Gao, Zhiyuan Yang, Junho Choi, Chan Wang, Guozhang Dai, Junliang Yang

**Affiliations:** 1School of Physics, Central South University, Changsha 410083, China; gao-mang@csu.edu.cn (M.G.); gzdai2011@csu.edu.cn (G.D.); 2Department of Mechanical Engineering, The University of Tokyo, Tokyo 113-8656, Japan; yang-zhiyuan@g.ecc.u-tokyo.ac.jp; 3Department of Mechanical Engineering, Tokyo City University, Tokyo 158-8557, Japan; choi@tcu.ac.jp; 4Department of Electrical and Computer Engineering, National University of Singapore, 4 Engineering Drive 3, Singapore 117576, Singapore; 5Center for Intelligent Sensors and MEMS, National University of Singapore, Block E6 #05-11, 5 Engineering Drive 1, Singapore 117608, Singapore

**Keywords:** triboelectric nanogenerators, preventive health monitoring, IoT, wearable electronics

## Abstract

With the improvement in life quality, the increased focus on health has expedited the rapid development of portable preventative-health-monitoring devices. As one of the most attractive sensing technologies, triboelectric nanogenerators (TENGs) are playing a more and more important role in wearable electronics, machinery condition monitoring, and Internet of Things (IoT) sensors. TENGs possess many advantages, such as ease of fabrication, cost-effectiveness, flexibility, material-selection variety, and the ability to collect low-frequency motion, offering a novel way to achieve health monitoring for human beings in various aspects. In this short review, we initially present the working modes of TENGs based on their applications in health monitoring. Subsequently, the applications of TENG-based preventive health monitoring are demonstrated for different abnormal conditions of human beings, including fall-down detection, respiration monitoring, fatigue monitoring, and arterial pulse monitoring for cardiovascular disease. Finally, the discussion summarizes the current limitations and future perspectives. This short review encapsulates the latest and most influential works on preventive health monitoring utilizing the triboelectric effect for human beings and provides hints and evidence for future research trends.

## 1. Introduction

In the modern world, people are paying more and more attention to health, driven by the aging global population and rising prevalence of age-related ailments. This has led to a growing demand for customized healthcare, particularly in the realm of preventive health monitoring [1]. Consequently, many portable electronic devices have been developed over the past decades to facilitate health condition monitoring, such as smart wristwatches, eyeglasses, shoes, bracelets, etc. [2]. In these portable electronics, sensing technologies play a significant role. As one of the most popular sensing strategies, triboelectric nanogenerators (TENGs) have attracted significant attention since their proposal in 2012 by Prof. Wang [3], owing to their self-powering ability, sustainability, low cost, and easy fabrication [4,5,6,7,8,9,10]. The application of TENGs has been expanded to and developed in various application domains, including blue energy harvesting [11,12,13,14], self-powered sensors [15,16,17,18], wearable devices [19,20,21], and IoT electronics [22,23].

Preventive health monitoring is an important component of healthcare for human beings because it can prevent diseases and injuries from happening or reduce side effects when minor symptoms occur [24]. With the development of IoT technologies, many portable electronics have been developed to achieve preventive health monitoring [25,26]. These electronics are normally equipped with sensors that can collect information from the human body or surrounding environment to assist people in avoiding possible dangers and diseases [27,28,29]. Therefore, sensing technologies are important for preventive health monitoring.

The triboelectric effect has been applied to healthcare electronics recently and is becoming increasingly pivotal, particularly with the growing prominence of preventive health monitoring [30,31,32]. Wang et al. [33] reviewed the progress and development of TENGs in healthcare, based on the different types of applications and sub-directions. Marjan Haghayegh et al. [34] reviewed wearable TENGs from the perspectives of the material selection, structure, and fabrication technique to further improve their shortcomings, such as limited comfortability and stretchability and a low current density. However, the overview of portable electronics based on TENGs to achieve preventive health monitoring is seldom discussed.

In this regard, the most recent progress in preventive health monitoring depending on triboelectric effects is introduced in this article. As shown in the schematic diagram in Figure 1, the main content is organized into three parts: the working principle of TENGs and the nanomaterials applied in them, applications of TENG-based devices for preventive health monitoring, and a discussion on the current limitations as well as future prospects. The section on the working principle and nanomaterials of TENGs applied in preventive health monitoring exemplifies the four modes of TENGs and gives some examples of the nanomaterials applied in them for preventive health monitoring. Subsequently, the application part introduces four scenarios of preventive health monitoring based on TENGs. The final segment delves into a discussion of the current limitations and prospects, offering guidance for future research.

## 2. Working Principle and Nanomaterials of TENGs Applied in Preventive Health Monitoring

The triboelectric phenomenon is an ancient phenomenon that has been known of for thousands of years. The mechanism was extended by Wang et al. in 2017 from the expression of the displacement current [35] and further illustrated as the first principle of TENGs from Maxwell’s equations in 2020 [36]. The triboelectric effect is the contact-induced electrification when a material comes into contact with a different material, based on which a TENG can convert mechanical energy into electrical energy after combining with the principle of electrostatic induction [37]. TENGs were invented with four working modes: the vertical contact–separation (CS) mode, lateral-sliding (LS) mode, single-electrode (SE) mode, and freestanding triboelectric-layer (FT) mode [38], as shown in Figure 2.

### 2.1. Contact–Separation Mode

The vertical CS mode employs relative motion that is perpendicular to the interface, where the potential change between the electrodes and external current flow is determined by the distance of the movement between the material surfaces [39]. A TENG, driven by airflow, was showcased for self-powered, real-time respiratory monitoring. The proposed device can achieve this by converting the mechanical energy generated by human respiration into electric output signals, and the basic working mode is the CS mode [40]. A wearable-triboelectric-sensor-enabled gait was fabricated for IoT-based, smart healthcare applications [41], and this smart device can achieve a novel fall detection system based on CS-mode TENGs. CS-mode TENGs have been widely utilized for injury monitoring [42], fall detection [43,44], respiration monitoring [45], biomedical monitoring [46], etc.

### 2.2. Lateral-Sliding Mode

The lateral-sliding mode involves the relative displacement parallel to the interface and can be compactly implemented through rotation-induced sliding [39]. Xu et al. [42] introduced an LS-mode TENG combined with a CS-mode TENG to distinguish motion signals from different body parts. This innovation has potential applications in post-rehabilitation training for patients with limb injuries and in smart medical care, offering effective information. Despite its potential, the LS mode has seen relatively limited application in healthcare domains.

### 2.3. Freestanding Triboelectric-Layer Mode

The freestanding mode evolves from the single-electrode mode, diverging by employing a pair of electrodes instead of the ground as the reference electrode. The electrical output is generated through asymmetric charge distribution as the freely moving object changes position [39]. Li et al. designed a self-powered smart wearable sensor (SWS) consisting of PTFE and Fe, which can be worn as a hand ring or foot ring for fall-down detection [47]. The fabricated devices based on the freestanding mode can be used as an alarm system for fall-down detection and sleep safety for babies, the elderly, and potential patients.

### 2.4. Single-Electrode Mode

The single-electrode mode (SE mode) of TENGs features a straightforward configuration in which an electrode is grounded. In this setup, contact–separation or sliding between triboelectric surfaces generates a current flow between the ground and the electrode [48]. This easy mode, with an electrode attached to only one triboelectric surface, makes SE-mode TENGs a preferred choice over paired-electrode TENGs. This characteristic facilitates their integration into various devices for self-powered sensing applications [39,48]. Wang et al. designed a shoes–floor TENG and applied it in fall detection [49] based on the single-electrode mode. The proposed shoes–floor TENG has broad application potential, including identification verification, posture calibration, and fall detection. To sum up, SE-mode TENGs are the most widely applied in the domain of preventive health monitoring.

### 2.5. Nanomaterials Applied in TENGs

It is well known that the output of TENGs mainly depends on the relative polarity of the chosen materials, and the polarities of many commonly used materials have been clearly described in the triboelectric series [50]. However, TENGs applied in preventive health monitoring require more advanced properties of used materials to achieve special functions, such as flexibility [51], ultra-high pressure sensitivity [51,52,53], specific gas detection abilities [54], and biocompatibility [55]. Nanomaterials with special surface treatments or synthesis methods are thus applied to meet these requirements.

For example, Kou et al. [51] fabricated a kind of porous PDMS by adding a mixture powder of fluorinated ethylene propylene (FEP) and citric acid monohydrate (CAM) into the PDMS solution. This porous nanostructure not only enabled substantial flexibility and durability but also increased the effective contact area as well as the pressure sensitivity of the device for sleep fall detection, as shown in Figure 3a.

Liu et al. [52] also mixed a PDMS solution with thermally expandable microspheres to create a pressure senor with ultra-high pressure sensitivity for wrist pulse monitoring. The microspheres show obvious deformation under small pressure, leading to the changing of the contact area. The number of microspheres increases with the mixing ratio, which gives a higher pressure sensitivity, as shown in Figure 3b.

In another work, Wang et al. [54] synthesized a Ce-doped ZnO-PANI nanocomposite film using the hydrothermal method and in situ polymerization process for ammonia sensing. The lower nanoparticles (Ce-doped ZnO) and the upper interconnected reticular nanofibers (PANI) can be clearly seen in Figure 3c, which increase the effective triboelectric area and NH_3_ adsorption sites at the same time.

Ouyang et al. [55] dissolved polylactic acid into chloroform and then further mixed in a Chiton solution, as displayed in Figure 3d. After drying and solidification, plasma etching was applied to provide a micro–nanostructure on the PLA/C film for a larger contact area. The as-fabricated film can achieve 99% sterilization and can totally degrade into H_2_O, CO_2_, and lactic acid when absorbed by the body, showing significant biocompatibility for implantable medical applications.

## 3. Applications of TENG-Based Preventive Health Monitoring

### 3.1. Fall Detection

The act of falling can be defined as an abrupt and unintentional shift in body position from a higher location to a lower one [40]. Statistics data indicate that approximately thirty percent of individuals aged 65 years and above fall every year [56]. The occurrences of falling can result in severe injuries, such as lacerations, cataclasis, or dislocations, as well as hematomas and contusions [57]. While most falls happen in bed [58], identifying such incidents in elderly individuals who are living alone or hospitalized, especially during the late night or early morning, presents a significant challenge. Due to the rapid aging of the global population, the significance of fall detection has risen significantly [59,60]. To address this problem, numerous methods for detecting falls have been suggested to mitigate the risk of further injuries associated with such incidents, including wearable-based [61,62,63] and vision-based [64,65] devices. However, these devices use extra sensors, like accelerometers and gyro-sensors, which need an external power supply. The vision-based methods use cameras to extract a change in body shape, which leads to high costs. Compared to these traditional methods, TENGs offer the benefits of self-powering ability and low cost simultaneously. Therefore, in the following sections, we will review related applications of fall monitoring based on TENGs.

In 2013, Jeon et al. [41] designed a fall detection system using a TENG array as a self-powered pressure-sensing component with an extremely simple structure. The TENG array consists of 15 thin-film, arch-shaped TENG-unit cells, and each cell incorporates PTFE and Kapton as the friction layers, while an Al foil works as the electrode (Figure 4a,b). Signal data over 48 normal daily activities and 48 falls were collected from eight participants and were then determined by the proposed fall detection system. It was found that the maximum number of activated cells (MAC) in a fixed window time (Figure 4c–f) showed a large difference between the daily activities and falls, and the MAC of the falls was much larger than that of the other daily activities (Figure 4g), which can be used as a representative feature for distinguishing daily activities from falls, achieving an accuracy of 95.75%, on average, with the window time set at 0.5 s, the training/test ration fixed at 56/40, and the MAC threshold set as 2.6.

In another work, Kou et al. [51] developed a smart pillow based on a flexible and breathable single-electrode-mode triboelectric nanogenerator (FB-TENG) sensor array, which can monitor head movements in real time during sleep. The FB-TENG consists of porous poly(dimethylsiloxane) (PDMS) fabricated by the sacrificial template method with citric acid monohydrate (CAM) and fluorinated ethylene propylene (FEP) powder, as shown in Figure 4h. It is worth noting that the CAM was first added and then removed by immersing it in ethanol to create a porous structure. By further combining this FB-TENG with a flexible printed circuit (FPC) and computer, a self-powered pressure sensor array (5 × 12) was achieved, which can realize the functions of pattern mapping (Figure 4i) and trajectory monitoring. When a person’s head rests on the smart pillow, the FB-TENG units positioned accordingly are activated. If the head comes into contact with any one of the five TENG pressure-sensing units situated at the outermost edge, the generated output signal will activate the alarm system, prompting the guardian or doctor to adjust the individual’s sleeping position and prevent the risk of falling out of bed, as revealed in Figure 4j.

In addition to monitoring falls from beds, there is the significant potential for sudden falls when elderly individuals venture out alone [58]. Consequently, there exists a significant need to develop wearable sensory systems that can offer real-time fall detection. A smart insole was developed by Lin et al. [66] in 2018 by introducing two single-electrode-mode TENGs based on copper and rubber friction layers in the front and rear of the insole, respectively. An elastic air chamber (EAC) part made of a designed elastic latex film is embedded underneath the TENG to promote the contact–separation by the airflow. The total design is sketched in Figure 5a. Human walking will apply force on the insole and achieve the contact and separation of the rubber and copper, which converts mechanical energy into electric signals. When the forefoot steps on the floor, a negative peak is generated from the front part of the insole, while a positive peak can be detected when the forefoot steps off the floor (Figure 5b). Because the output shows such corresponding peaks with the user’s gait, real-time fall detection can be achieved by integrating the signals of four smart insoles. If four positive peaks from the TENG-based sensors emerge at the same time, as shown in Figure 5c, this indicates a fall because both feet are off the ground. This study demonstrated a novel design in wearable devices for gait and fall monitoring.

After three years, Zhang et al. [44] reported a textile-based triboelectric sensory system with a single TENG sensor composed of five layers, where two TPU-coated polyester (PES)-fabric layers wrap the conductive nickel fabric (Ni-fabric), which simultaneously serves as the friction layer and electrode to maintain both the comfortability and flexibility. The negative triboelectric material is a thin silicone rubber layer with a pyramid-patterned surface created by 3D printing. A Ni fabric is coated on its backside and serves as the electrode (Figure 5d). This textile-sealing structure ensures the biocompatibility and durability of the device for at least 12 h after it is dipped in water, showing its great suitability for practical applications. After the arrangement of a couple of textile-based TENGs on the front and back sides of the insole, the human gait can be transferred into electric signals. When a sudden fall happens, the voltage signal appears as two negative peaks, with no more positive peaks thereafter, indicating that the user’s foot has left the ground, which triggers an alarm immediately (Figure 5e).

The aforementioned research has demonstrated the great feasibility of applying TENG-based devices for detecting abnormal falls to achieve preventive health monitoring. However, the usage of a great number of wires and computers in this research makes the whole system structurally clumsy and thus restricts real applications. Moreover, the failure to promptly notify family members or hospital staff is the primary factor contributing to the additional complications and potential fatalities resulting from falls [67]. Therefore, developing a wireless, portable human–machine interface (HMI) to remotely identify the human status and issue alerts regarding physiological signal changes is crucial for enabling real-time responses to emergent situations, particularly in cases of accidental falls. Wang et al. [49] proposed a simple human-motion-sensing system that can recognize human motion through a shoes–ground TENG, with a conductive hydrogel as the electrode for signal collection fabricated by a semi-interpenetrating polymer network method (Figure 5f). After assembling this conductive hydrogel in the human body, the shoes, floor, and human body are integrated into a single-electrode-mode TENG in which the shoes and floor work as the friction layers while the hydrogel serves as an electrode. This untraditional TENG can generate a maximum output of 700 V for shoes with rubber soles contacting common floor materials and remain constant despite the changing position of the hydrogel at the body. A remarkable sharp drop can be detected by shoes–floor TENGs when there is a fall happening, as displayed in Figure 5g. In addition, a wireless, mobile motion-monitoring system was further designed in this work. The shoe-based TENG was subsequently connected with a signal-conditioning circuit, an analog-to-digital converter, a microcontroller unit, and Bluetooth for wirelessly receiving and transferring the TENG signals from and to a cell phone without any signal losses. When the received signals exceed the threshold value, an alert message is automatically sent to the caregivers, as revealed in Figure 5h.

In another work, Li et al. [47] developed a self-powered smart wearable sensor (SWS) composed of PTFE and an Fe ball. The Fe ball can rotate on the side PTFE channel or strike the PTFE on the top and bottom channels to generate electric signals (Figure 5i). This sensor can be easily attached to the human body or clothing for condition monitoring. A cell phone is used to receive signals from the SWS wirelessly. When the SWS stops to generate signals, which indicates the falling condition, an emergency call is triggered automatically with the help of a smartphone app, as shown in Figure 5j. Additionally, TENG sensors can also be applied to the development of wearable sensors for fall detection with privacy protection at the same time [68,69].

Besides shoes and insoles, walking sticks, which serve as mobility aids for the elderly and those with limited motion, can improve the quality of their daily lives by enhancing their walking patterns, balance, and safety [70]. In addition to these fundamental functions, walking sticks, when combined with advanced wearable electronics, also have the potential to evolve into monitoring platforms, offering users more comprehensive and intelligent assistance [71]. Guo et al. [43] designed an AI-enabled caregiving walking stick equipped with intelligent monitoring features, and this device serves as a healthcare platform specifically tailored for the elderly and individuals with impaired mobility. A top press TENG (P-TENG) (Figure 6a(ii)) consisting of patterned Ecoflex and Nitrile as the friction layers, a rotational electromagnetic energy generator (EMG) (Figure 6a(iii)), and a rotational TENG (R-TENG) (Figure 6a(iv)) made of aluminum fans and PTFE are integrated into a unit to form a linear-to-rotary structure, as shown in Figure 6a. Five electrodes are then attached beneath the P-TENG. Nine typical contact points with their corresponding outputs are drawn in Figure 6b. By further combining MCU (Figure 6c) and AI deep learning to train the five-channel signal data from P-TENGs, multiple advanced functionalities, including mobility disability evaluation and motion status determination, were achieved, as displayed in Figure 6d. Moreover, the R-TENG is applied to detect falls. During regular movement, the R-TENG produces high outputs characterized by continuous high-speed rotation. However, in the event of a fall, irregular output patterns with low intensity can be detected, as illustrated in Figure 6e.

### 3.2. Respiration Monitoring

Respiration is a fundamental biomechanical process that persists throughout an individual’s entire lifespan and contains abundant physiological information [72,73]. In comparison to methods such as blood analysis and endoscopy, respiratory analysis emerged as a rapid, noninvasive, painless, cost-effective, and convenient approach for early illness diagnosis and real-time physiological monitoring [74,75,76]. The evaluation of the diaphragmatic movement during breathing facilitates the early detection of various human diseases, including conditions like apnea, asthma, cardiac arrest, and even lung cancer [77,78], which is crucial for preventive health monitoring and can serve as an effective method in preventive healthcare practices. Moreover, a novel coronavirus was identified in 2019 and is currently recognized as the severe acute respiratory syndrome coronavirus 2 (SARS-CoV-2) [79]. Given the strong contagious and harmful consequences, the early detection of COVID-19 plays an essential role in stopping its widespread transmission and protecting global public health security, which is urgent and in high demand all around the world [80].

In recent decades, several techniques have demonstrated promising outcomes in the analysis of the respiratory function, including ion mobility spectrometry (IMS) [81,82], gas chromatography [83], differential mobility spectroscopy [84], and thin-film transistors (TFTs) [85,86]. However, the majority of these techniques require an external power source to drive the sensors. Respiration, encompassing actions like the chest movement and airflow during breathing, represents a natural, periodic, and uninterrupted mechanical motion in humans. This motion can be applied as a consistent source of biokinetic energy to power low-energy wearable electronics and integrated body sensor networks [87], making TENGs an ideal and reasonable candidate for energy acquisition and the monitoring of respiration. In the following section, we review related applications of abnormal-respiration monitoring based on TENGs.

A respiration-driven TENG prepared by Ti_3_C_2_T_x_MXene/NH_2_-MWCNTs and nylon was developed for self-powered disease diagnosis by Wang et al. [88]. The device consists of a PVC tube, Ti_3_C_2_T_x_MXene/NH_2_-MWCNTs, and nylon. One end of the nylon film is fixed in the middle of the TENG (Figure 7a). Note that the Ti_3_C_2_T_x_MXene/NH_2_-MWCNT film works as a negative layer and electrode at the same time. The exhaled gas flowing through the PVC tube causes vibrations in the freestanding flexible nylon membrane, leading to repeated contact and separation with the Ti_3_C_2_T_x_MXene/NH_2_-MWCNTs. This process converts the mechanical energy from the breathing airflow into electric signals, which are transmitted to a computer for analysis, as shown in Figure 7b. In Figure 7c, the Cheyne–Stokes breath pattern is depicted, where the intensity of breathing gradually increases and then decreases, followed by a period of apnea often associated with heart disease [89]. Figure 7d illustrates the Cheyne–Stokes variant breath behavior, characterized by a period of low-intensity breathing, often indicating brain-stem lesions and cerebral-hemisphere lesions. Biot breathing, shown in Figure 7e, is distinguished by a period of breathing at a consistent intensity followed by a suffocation phase typically caused by brain-stem strokes and narcotic drugs [90]. To enable self-powered, real-time respiratory monitoring, an airflow-driven triboelectric nanogenerator (TENG) was demonstrated, converting the mechanical energy of human respiration into electric output signals [91]. By utilizing the TENG device, an advanced wireless respiratory monitoring and alert system was subsequently created. This system leverages the TENG signal to immediately activate a wireless alarm or dial a mobile phone, ensuring prompt notifications in the event of changes in breathing behavior.

Lu et al. [92] designed a novel, structured respiratory-sensing triboelectric nanogenerator (RS-TENG) for real-time respiration monitoring. An ultrathin FEP film and Al foil work as the triboelectric layers, with conductive cloth tape chosen as the electrode. Two acrylic boards are used as the substrate, with several holes drilled to allow the airflow to pass through. The breathing airflow can provide mechanical energy for the contact–separation of the RS-TENG and can thus generate electric signals, which are subsequently delivered to a single-chip microcomputer (SCM) for the wireless switching on/off of household appliances, such as a table lamps, by deliberate breathing, as shown in Figure 7f. Subsequently, an apnea alarm system was assembled by integrating it with a warning light, ensuring a prompt alarm when individuals cease breathing, as shown in Figure 7g.

In another work by Wang et al. [45], an eco-friendly, all-fabric TENG (AF-TENG) was successfully achieved based on both cotton fabric and ultra-high-molecular-weight polyethylene adhered with copper tape. This AF-TENG can be easily attached to a mask due to its light weight and good flexibility and can thus change the breathing airflow into electrical signals for real-time respiratory monitoring. An alarm system was developed by further connecting the mask with an Arduino Uno. When the voltage generated by the TENG overpasses the pre-determined threshold repeatedly, the display shows the message “BREATHING OK”, as shown in Figure 7h. If the voltage amplitude of the TENG cannot go beyond the threshold during a period, the display of the alarm system then changes from “BREATHING OK” to “NOT BREATHING” and the alarm starts to beep. Moreover, another IoT-based system (Figure 7i) was achieved for remote monitoring from a maximum distance of 20 km by linking the AF-TENG with a LoRa emitter, sending the voltage pulses generated by the TENG to a LoRa receptor operating as a gateway, which was connected by Wi-Fi to the internet. As a result, the breathing rhythm can be remotely monitored due to the inhalation and exhalation motion produced in the mask.

Aside from the breathing airflow, the chest or diaphragm muscle movement during breathing can be also utilized as an external energy source for breathing analysis. Ning et al. [93] developed a helical fiber strain sensor (HFSS) by incorporating a helical structure into a stretchable substrate fiber. The HFSS was created by alternately winding PTFE fibers and nylon fibers around the stretchable substrate fiber. The electrical signals can be generated through the contact–separation of the two triboelectric layers, achieved by the stretching and releasing motions of the HFSS, as shown in Figure 7j,k. Owing to the well-designed structure and proper material selection, HFSSs show high sensitivity to strain, and they can generate a voltage of 0.5 V even under a 1% stretch strain. The expansion and contraction of the chest circumference during respiration lead to the contact and separation of the PTFE and nylon fibers. Real-time electric output profiles reflecting breathing patterns for expiration and inspiration can thus be observed, as shown in Figure 7l. In addition, a self-powered, intelligent alarm based on HFSSs was further developed (Figure 7m) by connecting different electronic components and smartphones. If the breathing signal halts for more than 6 s, the alarm system will automatically call the preset mobile phone for assistance.

Peng et al. [94] introduced a self-powered all-nanofiber e-skin (SANES) for continuous respiratory monitoring, diagnosis, and preemptive measures against obstructive respiratory diseases by electrospinning, wherein electrospun polyamide 66 (PA 66) nanofibers coated with a Au electrode serve as the top electrification layer, and polyacrylonitrile (PAN) nanofibers, also coated with a Au electrode, form the bottom electrification layer, as shown in Figure 7n. This all-nanofiber design allows this device high sensitivity (0.217 kPa^−1^), good flexibility, wearing comfort, and convenience. Notice that the interlaced nanofibers create numerous spatial hierarchical, porous micro-to-nanostructures. These structures contribute to a high specific surface area for contact electrification and boost the power output of the SANES with a maximum power density of 330 mV/m^2^. By attaching the SANES to the abdomen by a medical bandage, the diaphragm muscle pulls down and up during the inhalation and exhalation, which provides mechanical energy for the contact and separation of the SANES and thereby delivers an AC electric output. Therefore, the respiratory rate and patterns can be precisely and spontaneously assessed and recorded, as shown in Figure 7o. In addition, an intelligent system for obstructive sleep apnea–hypopnea syndrome (OSAHS) diagnosis was built up based on the as-prepared breath analyzer by further connecting it with a signal-processing circuit and computer. Real-time voltage signals under different sleep respiratory states (normal, hypopnea, apnea) can be clearly detected, as displayed in Figure 7p–r.

In addition to directly detecting physical respiratory patterns, respiration also plays a pivotal role in the exchange of air between the lung’s alveolar spaces and the surrounding atmosphere. The exhaled gases comprise a complex mixture of up to 500 different chemical compounds, including carbon dioxide, nitrogen, oxygen, water vapor, and volatile organic components [95,96,97]. These constituents and their concentrations can serve as biomarkers, offering a wealth of biochemical and physiological information. For example, ammonia is the biomarker for kidney disease [98], ethanol can be employed as an indicator for liver cirrhosis [99], and the acetone concentration in exhalation gas can be regarded as an indicator that reflects prediabetes [100]. This information can be priceless for disease diagnosis and early intervention [101]. In the following part, the TENG-based sensor for respiratory chemical regent detection is reviewed.

Aside from the pre-mentioned indicator for liver cirrhosis, alcohol-related transportation accidents represent a significant public safety issue. In the United States, alcohol is a contributing factor in more than 30% of all traffic fatalities [102]. Wen et al. [103] devised a self-powered ethanol sensor using a blow-driven triboelectric nanogenerator (BD-TENG). This innovative sensor consists of a sliding-mode (rotation) TENG with nanowire-patterned fluorinated ethylene propylene (FEP) working as the negative friction layer and copper playing the roles of the positive friction layer and electrode simultaneously (Figure 8a). A gas-sensing, rhombus-shaped Co_3_O_4_ nanorod array prepared via a two-step fluorine-mediated hydrothermal process is then connected with the TENG in series to form the sensor system. The exhaled breathing gas flow drives the friction between the fluorinated ethylene propylene (FEP) and copper to generate electric outputs. It should be noted that the generated voltage is proportional to the impedance of the whole system, which is associated with the chemisorption process of Co_3_O_4_ in response to the target exhaled gas. In vacuum, the primary charge carriers in Co_3_O_4_ are holes. Upon exposure to air, the surface of Co_3_O_4_ quickly becomes covered with negatively charged chemisorbed oxygen species. Consequently, an increase in conductivity occurs due to the generation of holes and the formation of a charge accumulation layer on the surface. However, after introducing alcohol, the accumulation layer is thinned through electrochemical interactions between the negatively charged chemisorbed oxygen and ethanol molecules. This interaction releases free electrons, neutralizing the generated holes and thereby increasing the resistance. The whole process is shown in Figure 8b. Therefore, this Co_3_O_4_ sensing material can work as a rheostat and change the resistance of the whole system corresponding to the concentration of alcohol. The prepared TENG can generate higher output when the sensing material is exposed to an alcohol environment, as shown in Figure 8c, based on which a self-powered alcohol-sensing system was developed by integrating a signal-processing circuit for the rapid recognition of the drunkenness level, as revealed in Figure 8d.

Wang et al. [54] proposed a respiration-driven triboelectric ammonia sensor consisting of laser-patterned PDMS coated with a gold electrode and Ce-doped ZnO-PANI film fabricated by hydrothermal synthesis. Drawing inspiration from the kinetic features of the lung lobe, the as-proposed sensor is linked to an elastic airbag to capture human respiratory motions. This setup facilitates the contact and separation between the PDMS and Ce-doped ZnO-PANI film, leading to the generation of electrical signals, as depicted in Figure 8e. It should be noted that the Ce-doped ZnO-PANI film simultaneously works as a contacting layer, gas-sensing material, and electrode due to its excellent conduction and NH_3_-sensing abilities. The respiratory patterns of slow, rapid, and deep breathing can be explicitly distinguished in terms of the output signals based on this sensor, as shown in Figure 8f. Moreover, the prepared sensor shows decreasing output with an increasing NH_3_ concentration, which can be attributed to the decrease in the conductivity of the Ce-doped ZnO-PANI composite film after it is exposed to NH_3_ due to the electrons donated by NH_3_ to PANI during the chemical reaction. The sensor also exhibits very good linearity, up to 0.9928 between the NH_3_ concentration and Response, which is calculated as 100% × (V_g0_ − V_g1_)/V_g0_, where V_g0_ is the initial output voltage in air and V_g1_ is the measured voltage in the target gas, as displayed in Figure 7h and Figure 8g, suggesting the potential for pre-diagnosing kidney diseases. The sensor’s capability in an oral atmosphere was assessed by measuring the real-time output voltage patterns under various oral conditions, including brushing teeth, not brushing teeth, and having dental ulcers, as shown in Figure 8i. The ulcer patient generates a lower output voltage compared with the other two cases due to the inflammation in the mouth cavity, releasing more ammonia molecules.

Su et al. [100] developed a wireless-power-transmission-enabled self-powered acetone sensor (WSAS) consisting of chitosan and reduced graphene oxide (rGO) as the sensitive materials for prediabetes diagnosis. The prepared sensor is composed of two main parts: a power generation component and a gas-sensing component. The exhaled gas flow vibrates the PTFE film and gives rise to its contact–separation with the nylon layer, yielding outputs that can be collected by the copper electrode coated with the sensing layer. Notice that unlike conventional TENGs, the copper electrode in this sensor system is not adhered to either of the two friction layers but is coated on the bottom side of the acrylic sheet in the test chamber via thermal evaporation. The whole system is depicted in Figure 8j. In the test chamber, the chemisorption of acetone molecules alters the permittivity of the sensing layer, influencing the screening effect of the sensing layer and leading to a significant variation in the output signal. As shown in Figure 8k, the sensor ability was tested by analyzing exhaled gases 1 h after the ingestion of different amounts of anhydrous glucose. The increase in the glucose intake raises the acetone concentration in the exhaled breath and corresponds to the enhancement of the voltage amplitude, showing its capability to diagnose prediabetes. Moreover, the selectivity of the developed sensor was assessed by examining a range of harmful and toxic gases, as illustrated in Figure 8l. The theoretical gas-sensing mechanisms among nitrogen, oxygen, moisture, and acetone were also systematically investigated by phase-field simulation and finite element analysis, as shown in Figure 8m–p. These results indicate its promising acetone-sensing ability and feasibility, even under a complex gas atmosphere.

### 3.3. Fatigue Monitoring

According to the data released by the World Health Organization (WHO), approximately 1.2 million individuals succumb to traffic accidents globally each year [104,105]. Statistics reveal that, on average, one person loses their life in a traffic accident every 5 min, and someone becomes disabled due to a traffic accident every minute. Additionally, hundreds of billions of dollars in economic losses result from traffic accidents annually. Moreover, the data assert that a significant portion of these accidents can be attributed to driver fatigue and carelessness. The American Automobile Association reported that 7% of all accidents and 21% of fatal traffic accidents were caused by tired drivers [106]. In China, deaths resulting from traffic accidents rank second globally, with approximately 20% of car crashes attributed to fatigue driving [107]. Therefore, monitoring the driver’s status and issuing timely reminders about fatigue driving hold immense significance in reducing the occurrence of traffic accidents. Traditional methods for detecting fatigue driving mainly rely on computer vision [108,109] for detecting the behaviors of blinking, yawning, and head moving, which requires high-cost cameras and an external power supply. Fortunately, the behaviors of blinking, yawning, and head moving can also generate mechanical energy, which can be converted into electrical signals by TENGs, making them a promising candidate for fatigue detection. Herein, we will review TENGs as self-powered sensors for biological signal detection and driver status monitoring.

In 2018, Meng et al. [110] first introduced a TENG-based sensor to monitor driver behavior with a driving simulator. The above-mentioned TENG is composed of two pieces of Al foil adhered to Kapton, with another piece of Al placed in the middle of the TENG as the interlayer (Figure 9a). The fabricated TENG has very good pressure sensitivity and can be used as a self-powered sensor for monitoring the driver’s behavior. The monitoring system consists of three TENGs as sensors, with two TENG sensors mounted on the accelerator and brake of a driving simulator and the last one in contact with the corner of the driver’s eye. All three sensors are then linked to a multichannel data acquisition device with a laptop. The whole system is schematically illustrated in Figure 9b. The small changes in the eye muscles during blinking can provide mechanical energy for the contact–separation of the TENG and thereby generate signals (Figure 9c), which can reflect the blink frequency and duration, and this blinking information during the car-driving process can be used for fatigue warning. The pressing of the accelerator and brake by the feet can also be recorded by the attached TENGs, as shown in Figure 9c, which can reveal some bad driving behaviors, including impulse and drowsiness driving.

After two years, Lu et al. [111] demonstrated the use of a flexible, transparent single-electrode-mode triboelectric nanogenerator as a sensor to detect driver fatigue, which consists of ionic hydrogel electrodes made of polyacrylamide (PAAM) and LiCl sandwiched between two PDMS friction layers. This PAAM-LiCl-based triboelectric nanogenerator (PL-TENG) has the advantages of a small size, a light weight, high output, soft stretchability, high transparency, and good biocompatibility, and it can thus stick on the surface of human skin. The as-fabricated TENGs are then attached to the corners of the eyes, the mouth to the lower jaw, the back, and the left of the neck of a tester, as shown in Figure 9d. Based on these TENGs attached to the corresponding parts, the information on blinking, the yawning process, the head bowed down, and neck twisting can be monitored as electrical signals generated by the contact–separation between the skin and attached TENGs for analyzing the driving status. For example, the blinking duration under the normal driving condition is lower than that under the fatigue driving condition (Figure 9e), while the blinking interval duration (BID) shows the reverse tendency (Figure 9f). The yawning (Figure 9g,h) can directly reflect the fatigue of the driver. The PERCLOS, which refers to the percentage of time that the eyes are closed within a certain time window, can also be used for fatigue judging. A fatigue-judging criterion was further established by setting thresholds for the four mentioned parameters, as shown in Figure 9i. A buzzer is triggered when the blinking duration is more than 2 s to remind the fatigued driver.

In 2023, Luo et al. [112] introduced a flexible polyethylene oxide (PEO) polydimethylsiloxane (PDMS)-based triboelectric nanogenerator (PP-TENG) for detecting fatigue during driving through respiration monitoring. Through the incorporation of a sodium chloride solution into PEO and tea powder into PDMS, the sensor’s output voltage and current achieved significant increases of approximately 9.75 and 8.21 times, respectively. The sensor exhibited a sensitivity of 0.7 V/kPa within the linear range of 0–50 kPa and demonstrated a rapid response time of 36 ms. The schematic illustration of the structure of the PP-TENG is shown in Figure 9j. The as-fabricated sensor can be attached to people’s chests to collect the respiration signals generated by the expansion and contraction of the chest during the breathing process. Because people have different breathing states when they are tired or awake, respiration signals can be used to indicate the states of drivers. Four respiration-related parameters, the respiratory rate (RR), mean respiratory amplitude (MRA), standard deviation of the respiratory amplitude (σ_RA_), and yawn frequency (YF), are used for judging the levels of fatigue. A total of 36 groups of people were tested, and the values of the MRA, σ_RA_, and RR in the fatigued state were found to be less than those in the normal state (Figure 9k–m) due to the slow and gentle chest movement in the fatigue condition, leading to the reduction in the frequency, amplitude, and fluctuation of the output voltage. A fatigue-judging criterion was then established based on these parameters, as shown in Figure 9n. Moreover, some other fatigue-monitoring methods with TENGs on wheels have also been reported [113,114].

### 3.4. Preventive Monitoring for Cardiovascular Disease

In contemporary healthcare, the arterial pulse serves as a crucial indicator for assessing the arterial blood pressure and heart rate, offering valuable insights for noninvasive medical diagnoses and preventive health monitoring. In the initial stages of certain conditions, like asymptomatic cardiovascular diseases, such as atherosclerosis, the arterial pulse may exhibit pathologic changes, impacting the arterial blood pressure. Consequently, the ongoing monitoring of the arterial blood pressure through wrist pulses offers a swift and noninvasive method for the early diagnosis of cardiovascular diseases [115]. In the following part, we will review TENGs as self-powered sensors for arterial pulse monitoring.

Wu et al. [46] report a multi-mode stretchable triboelectric nanogenerator (msw-TENG) based on liquid metal and silicone material for wrist pulse monitoring. The liquid metal EGaIn (eutectic gallium–indium alloy) is applied as a stretchable electrode due to its high electrical conductivity, low viscosity, deformability, and non-toxicity. The liquid metal is sandwiched between two silicone plates in a well-designed island–bridge structure created through 3D printing, as shown in Figure 10a. Upon the applied external force to the center of the msw-TENG, the deformation of the silicone results in the squeezing of the liquid metal to both sides through narrow channels, causing the liquid metal on each side to effectively bulge the silicone, as shown in Figure 10b, leading to a substantial alteration in the effective contact area between the liquid metal and the silicone, which generates electrical signals. The specific working principle is displayed in Figure 10c. This ingenious island–bridge structure enables the msw-TENG ultra-sensitive response to small presses, and the msw-TENG can be used as a wearable active sensor for detecting the arterial pulse. The pulse information, including the pulse amplitude and frequency, under three different physiological conditions can be clearly detected, as shown in Figure 10d.

Liu et al. [52] achieved high-sensitivity, self-powered, and flexible-pressure TENG sensors created by combining PDMS and thermally expandable microspheres. The PDMS agent mixed with different ratios of microspheres is spin-coated onto a copper electrode to form a contact–separation-mode TENG, as displayed in Figure 10e. After they are heated, the microspheres expand and appear on the original flat PDMS surface, which enables the high pressure sensitivity of the sensor, reaching a maximum sensitivity of 150 mV/Pa at a weight percentage of 1%. The as-proposed pressure sensor was put on the radial artery of the wrist for pulse monitoring, as shown in Figure 10f, and a pulse rate of 100 beats/minute was clearly recorded. Furthermore, the pulse signal distinctly displays the characteristic peaks found in peripheral artery waveforms, including the pulse pressure (P-wave) and late systolic augmentation (D-wave), as displayed in Figure 10g, which encompass vital biomedical and physiological information that is crucial for diagnosing cardiovascular diseases [116,117].

In another work, Wang et al. [53] proposed a simple method for fabricating large-area, patterned PDMS thin films by using silk as the molds. The mixture of the PDMS base and crosslinker is directly poured onto the surface of a piece of clean silk scarf and then peeled off from the silk after solidifying, as shown in Figure 10h. The TENG-based e-skin consisting of the textile-patterned PDMS shows an significant sensitive response to pressure with an extremely low detection limit of 0.6 Pa, which can give a fast response to even small insects, like ants (10 mg) and bees (40 mg), as shown in Figure 10i. Due the extraordinary pressure sensitivity, the e-skin can be placed over the radial artery of the wrist to distinguish the subtle pressure difference between people with different body conditions. Figure 10j presents the real-time pulse signals of the e-skin device over the artery of a wrist for a healthy person and a pregnant woman, and it clearly shows that the pulse frequency of a pregnant woman is faster than that of a healthy person.

In addition to indirect monitoring methods for cardiovascular disease, such as pulse detection on radial arteries [118], some direct methods are also proposed based on TENGs. Zheng et al. [119] designed an innovative implantable triboelectric nanogenerator (iTENG) for in vivo biomechanical energy harvesting and real-time wireless cardiac monitoring, as shown in Figure 11a. The power source was the heartbeat of an adult Yorkshire porcine, and the iTENG worked for over 72 h of implantation in the active animal with a relatively excellent performance. Various implantation sites for optimal energy harvesting were evaluated, and the signal collected from the iTENG was different when epinephrine was administered. This work shows great potential for fabricating a self-powered, wireless healthcare-monitoring system. Moreover, TENGs hold the promise of prolonging the device operation time within the body and mitigating the need for high-risk repetitive surgeries, as seen in the case of pacemakers. To this end, a high-performance inertia-driven triboelectric nanogenerator (I-TENG) of a commercial coin battery size was developed, harnessing the energy derived from body motion and gravity [120] (Figure 11b). The proposed device has successfully demonstrated the ability to charge a lithium-ion battery and can be used for data monitoring via Bluetooth with the real-time output voltage. The device was packaged with the commercial medical-purpose biocompatible polymer to verify the biocompatibility, and there was no sign of infection during the observation period. Additionally, the cardiac pacemaker system can be integrated with the I-TENG, confirming the ventricle pacing and sensing operation mode of the self-rechargeable cardiac pacemaker system. The designed device shows potential for the development of new self-rechargeable, implantable medical devices with health-monitoring capabilities.

Ouyang et al. [121] proposed a flexible self-powered ultrasensitive pulse sensor (SUPS) based on a triboelectric active sensor consisting of nanostructured Kapton and copper, which can provide the accurate, wireless, and real-time monitoring of physiological information for cardiovascular diseases based on pulse signals, as shown in Figure 11c. The as-designed SUPS can detect the variation in pulse waves under three different working modes of the implanted intra-aortic balloon pump (IABP) and distinguish the systole and diastole of the heart when the cardiac load or vascular pressure changes, showing great potential for intelligent cardiovascular disease diagnosis and prevention. In 2021, the same research group, Ouyang et al. [55], presented an implantable, bioresorbable, self-powered sensor consisting of the antibacterial bioresorbable materials poly(lactic acid)–(chitosan 4%) (PLA/C). Notably, the sensor can be absorbed directly into the organism after its service life, serving as an integral part of the body. For showcasing its capability for identifying abnormal cardiovascular events, the sensor was affixed to the vascular wall of a dog and electrical signals were obtained from the flowing blood, which successfully detected the abnormal arrhythmia and vascular occlusion, as shown in Figure 11d, showing great potential for the prognosis of cardiovascular surgery.

## 4. Discussion and Prospects

In response to the rising healthcare expenses and shortage of nursing staff in our aging society, there is a shift towards transferring the care system to home environments whenever feasible. This transition leverages ambient assisted-living technology, giving high requirements of sensing technologies for preventive health monitoring. In this review, the most recent preventive-health-monitoring technologies based on triboelectric effects are summarized by introducing five aspects of related applications, including fall detection, respiration monitoring, fatigue monitoring, pulse monitoring, and monitoring for cardiovascular disease. Some typical application functions and their applied materials are summarized in Table 1.

The current limitations and possible prospects, which can be further expanded, are summarized as follows:TENG-based sensors have many advantages when applied in fall detection for aging and disabled people due to their easy fabrication, low cost, real-time-monitoring ability, privacy proception, etc. Moreover, they can be easily integrated into a cane or shoes [71,122,123] to achieve the self-powered ability, which has the potential to eliminate the limitations of charging problems, giving aging and disabled people more convenience. In addition, they can be also applied to large healthcare centers and combined with carpets, which can protect the privacy of aging and disabled people [124]. However, according to the current research on TENG sensor systems in preventive health monitoring, they still need outside power sources, especially for ancillary systems, such as signal collection systems, signal transmission systems, and human–computer interaction devices. Therefore, the control circuit still needs to be further customized to make it suitable to the characteristics of TENGs, and low-power electronics should also be developed to cooperate with TENG-based sensors to achieve completely self-powered health-monitoring devices;The respiration-monitoring technologies based on TENGs can give a good early warning for many diseases, such as abnormalities in apnea, asthma, cardiac arrest, and even lung cancer, thereby leaving users and medical workers with more time to prevent the severity of the diseases. However, for real applications, portable TENG-based monitoring care systems for home environments are pressing. In addition to this, for clinical applications, a comparison of the research with that of medical researchers should also be conducted to achieve more accurate and practical applications;TENG-based fatigue monitoring has great potential for monitoring the driver’s status to avoid accidents and injuries and has the advantage of protecting the privacy of drivers compared with camera monitoring technologies. However, the response time is still very long for driving conditions [111], and thus a quick-response TENG should be developed, as accidents can happen in less than 1 s. Most of the fatigue monitoring based on triboelectric effects is used for driving monitoring, but it also has the potential to be applied in other dangerous-operation positions, such as high-altitude platform work;TENG-based pulse and cardiovascular monitoring is a very easy method for the early diagnosis of cardiovascular disease. However, because this kind of application requires that the TENG sensors are directly attached to the human body or even implanted inside of it, the biocompatibility of the materials used is crucial. Furthermore, because a long service life is always expected, some new materials with robust durability that can work stably in liquid environments (sweat, blood) for real applications need to be developed;Currently, most preventive-health-monitoring technologies are based on polymer materials, and the long-term durability is still a critical issue for practical applications. Some non-polymer materials can also be used as preventive-condition-monitoring materials that have good biocompatible abilities and high durability, such as DLC and Si-DLC [125,126];Preventive health monitoring will play a more and more important role in our daily lives in an aging society, as preventive health monitoring has the potential to reduce healthcare fees and prevent the severity of diseases and possible injuries and accidents. The flexible working mode and plentiful material selection of TENGs provide more possibilities for future preventive-health-monitoring sensing systems with the help of IoT and AI technologies. Currently, the research on TENG-based preventive health monitoring is still in the laboratory stage, and practical exploration cooperating with medicine and computer science should be on the agenda to make it smaller and more accurate.

## Figures and Tables

**Figure 1 nanomaterials-14-00336-f001:**
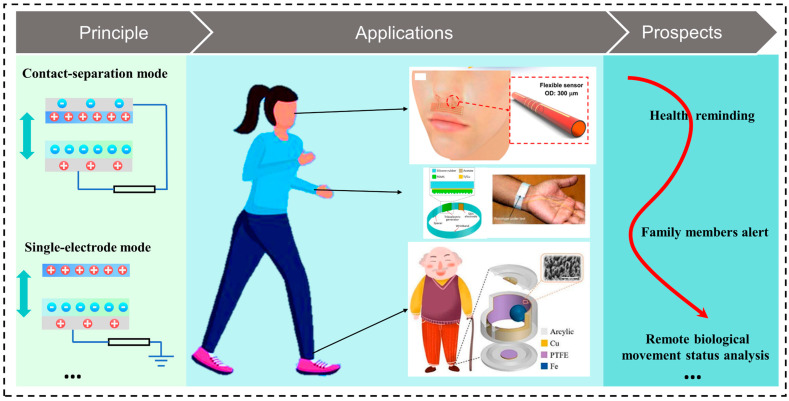
Diagram of overview of this paper.

**Figure 2 nanomaterials-14-00336-f002:**
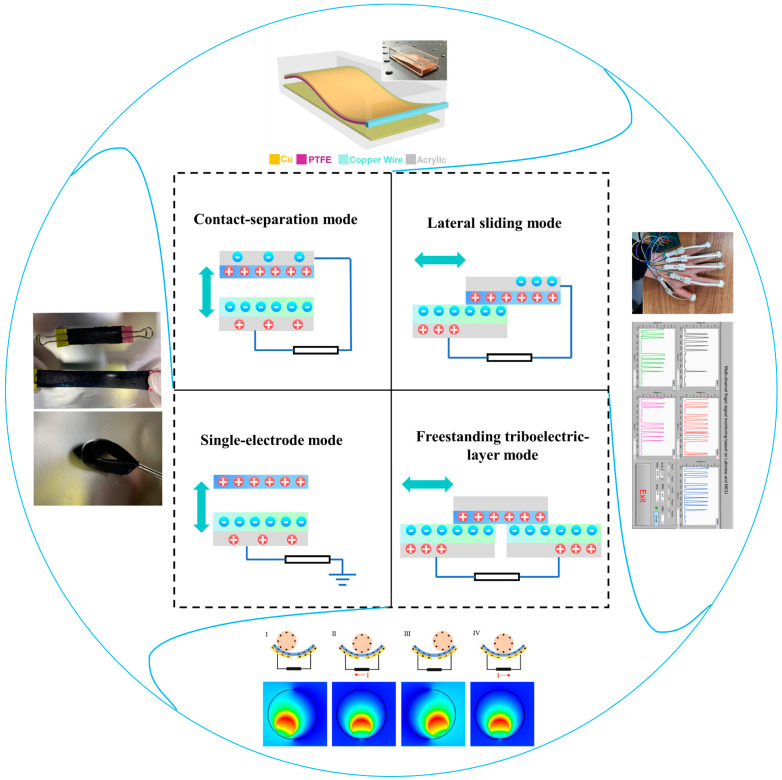
Four working modes of TENGs and their application in preventive health monitoring.

**Figure 3 nanomaterials-14-00336-f003:**
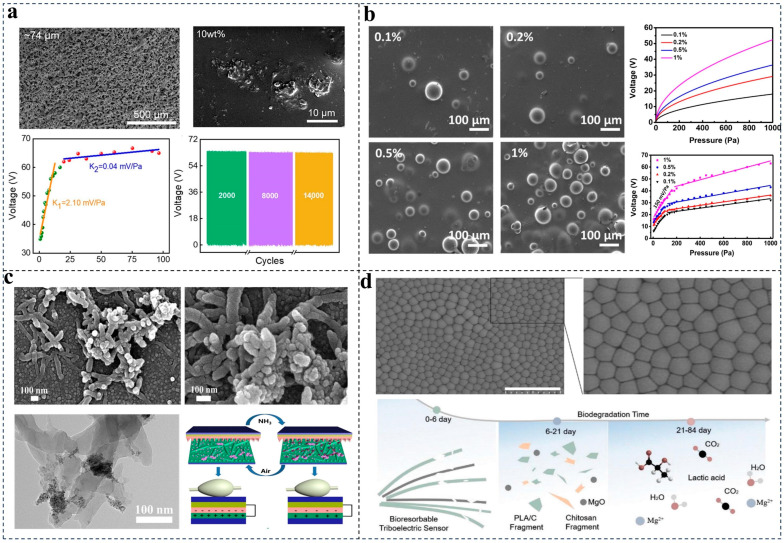
Nanomaterials applied in TENGs. (**a**) Porous PDMS made with 74 ± 6 μm CAM particles and 10 wt % FEP powder for smart pillow. Reproduced with permission [51]. Copyright 2022, American Chemical Society. (**b**) PDMS–microsphere layer with microsphere mixing ratios of 0.1%, 0.2%, 0.5%, and 1% for pulse monitoring. Reproduced with permission [52]. Copyright 2019, Elsevier. (**c**) Ce-doped ZnO-PANI sensitive film for ammonia detection. Reproduced with permission [54]. Copyright 2019, Elsevier. (**d**) Antibacterial bioresorbable materials poly(lactic acid)–(chitosan 4%) (PLA/C) for implantable cardiovascular postoperative care. Reproduced with permission [55]. Copyright 2021, Wiley-VCH.

**Figure 4 nanomaterials-14-00336-f004:**
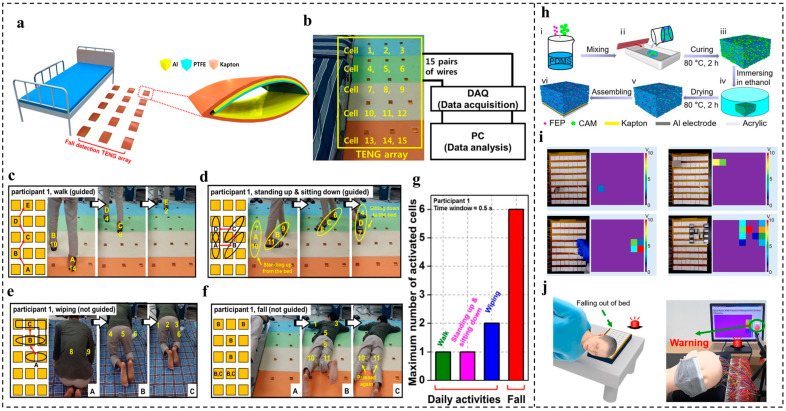
TENG-based fall detection system. (**a**) Schematic of the pressure-sensing TENG array and TENG-unit cell. (**b**) Photograph of the TENG array and system schematic of the proposed fall detection system. Trajectory and photographs of (**c**) walking, (**d**) standing up and sitting down, (**e**) wiping, (**f**) falling. (**g**) Extracted maximum numbers of activated cells in daily activities and falls. Reproduced with permission [41]. Copyright 2013, Elsevier. (**h**) Schematic diagram of the fabrication process of the porous PDMS and the FB-TENG in single-electrode mode. (**i**) Application of the FB-TENG array in pattern mapping. (**j**) Schematic diagram when a person is in danger of falling out of bed. Reproduced with permission [51]. Copyright 2022, American Chemical Society.

**Figure 5 nanomaterials-14-00336-f005:**
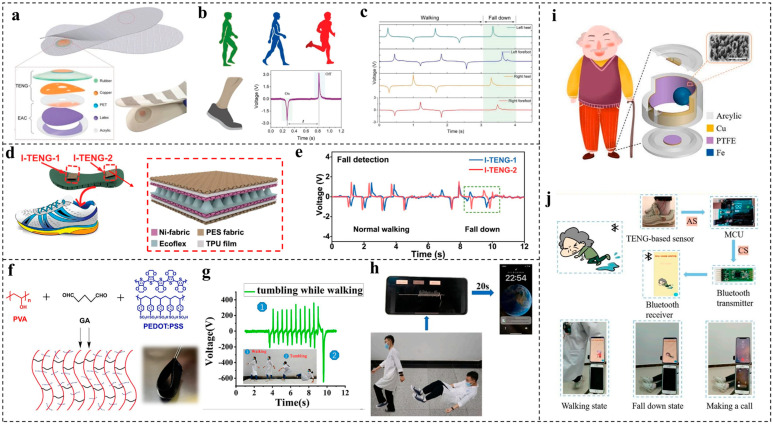
TENG-based wearable devices for fall detection. (**a**) The smart insoles assembled into shoes. (**b**) Typical detected signals when the footstep is on and off. (**c**) Sensing signals for walking. Reproduced with permission [66]. Copyright 2018, Wiley-VCH. (**d**) Textile-based TENG smart insole. (**e**) Signal generation for detecting falls. Reproduced with permission [44]. Copyright 2019, American Association for the Advancement of Science. (**f**) Synthesis mechanism of the SIPN PVA/PEDOT: PSS hydrogel. (**g**) Signal generation of walking with a sudden fall. (**h**) Demonstration of the wireless, mobile motion-monitoring system. Reproduced with permission [49]. Copyright 2023, Elsevier. (**i**) Structure of SWS attached to shoe for fall detection. (**j**) Demonstration of SWS for motion state monitoring and fall-down alarm based on smartphone app [47].

**Figure 6 nanomaterials-14-00336-f006:**
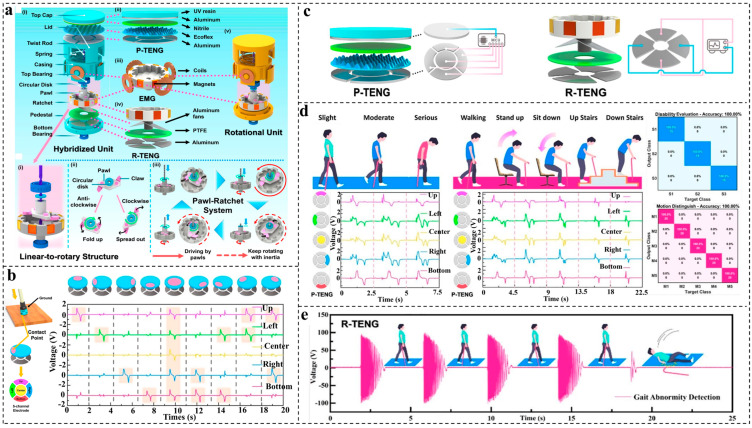
Multifunctional monitoring system enabled by the caregiving walking stick. (**a**) Schematics of the structure of the caregiving walking stick. (**b**) Schematic drawing of the contact point and five bottom electrodes labeled with Up, Left, Center, Right, and Bottom and their corresponding outputs with nine typical contacts. (**c**) The data collection of P-TENG and R-TENG by MCU. (**d**) Mobility disability evaluation and motion status determination achieved by the smart caregiving walking stick. (**e**) Output curve for the R-TENG under normal walking status and falling. Reproduced with permission [43]. Copyright 2021, American Chemical Society.

**Figure 7 nanomaterials-14-00336-f007:**
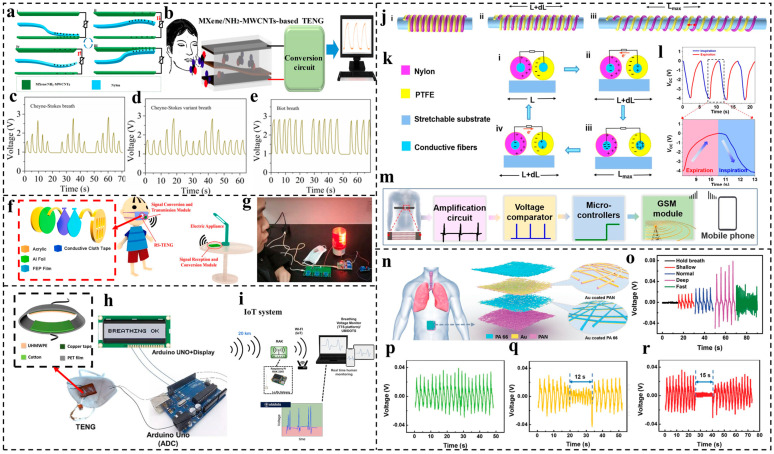
TENG-based wearable devices for respiration monitoring. (**a**) The working mechanism of the MXene/NH_2_-MWCNT-based TENG. (**b**) Application diagram of MXene/NH_2_-MWCNT-based TENG. Respiration signal patterns for (**c**) Cheyne–Stokes breath, (**d**) Cheyne–Stokes variant breath, (**e**) Biot breath. Reproduced with permission [88]. Copyright 2022, Elsevier. (**f**) Structure of RS-TENG and related control circuit diagram of smart facemask. (**g**) The triggered alarm after the smart facemask is removed (breathing stop). Reproduced with permission [92]. Copyright 2022, Elsevier. (**h**) Structure of the AF-TENG and related alarm system. (**i**) Electronic and communication setup of remote breathing monitoring using the LoRa protocol. Reproduced with permission [45]. Copyright 2023, American Chemical Society. (**j**) Structural illustrations of HFSSs in different states. (**k**) Working principle of HFSSs. (**l**) Regular electrical signals of the HFSS-based chest strap as the human body breathes. (**m**) Schematic diagram of a real-time respiratory monitoring and intelligent alert system. Reproduced with permission [93]. Copyright 2022, American Chemical Society. (**n**) Structural design and working principle of the TENG-based SANES. (**o**) Real-time voltage signals of different respiratory states. Real-time voltage signals under different sleep respiratory states, including (**p**) normal, (**q**) hypopnea, and (**r**) apnea. Reproduced with permission [94]. Copyright 2021, Wiley-VCH.

**Figure 8 nanomaterials-14-00336-f008:**
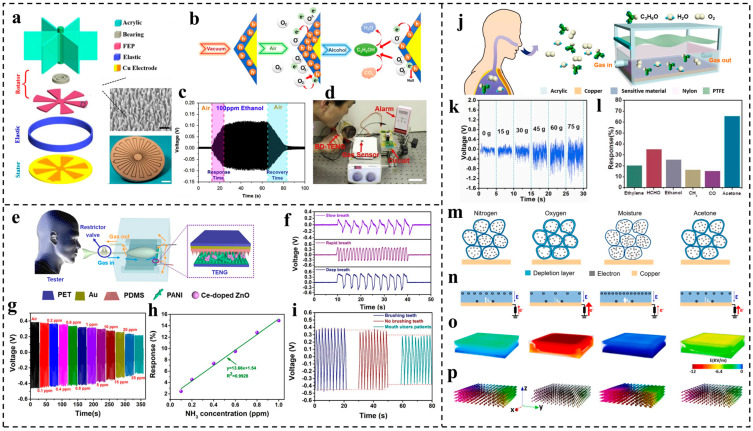
TENG-based sensor for respiratory chemical regent detection. (**a**) Device structure of the blow-driven TENG (BD-TENG). (**b**) Schematic diagram of the chemical reactions under the applied voltage generated by the BD-TENG for self-powered gas sensing. (**c**) Dynamic voltage response of the BD-TENG to ambient alcohol concentrations. (**d**) Photograph showing the BD-TENG acting as a self-powered breath analyzer. Reproduced with permission [103]. Copyright 2015, Elsevier. (**e**) Schematic illustration of human respiration-driven system. (**f**) Real-time respiratory signals of the TENG under three different human breathing behaviors. (**g**) Output voltages detected at different NH_3_ concentrations from 0.1 to 25 ppm at a fixed respiratory flow. (**h**) Response to NH_3_ concentration fitting curve of the self-powered respiratory sensor at 0.1–1 ppm NH_3_ atmosphere. (**i**) Comparison of output voltages under different oral environments at a fixed respiratory flow. Reproduced with permission [54]. Copyright 2019, Elsevier. (**j**) Structural design and output performance of the wireless-transmission-enabled self-powered acetone sensor (WSAS). (**k**) Real-time monitoring of the exhaled acetone after 1 h of anhydrous glucose intake. (**l**) The selectivity of the as-fabricated gas sensor. Schematics of (**m**) chemical adsorption processing and (**n**) charge alignment under an external electric field. Distribution of (**o**) electric field and (**p**) polarization in the sensing layer. Cases of measurements toward different gases, including nitrogen (1st column), oxygen (2nd column), moisture (3rd column), and acetone (4th column). Reproduced with permission [100]. Copyright 2020, Elsevier.

**Figure 9 nanomaterials-14-00336-f009:**
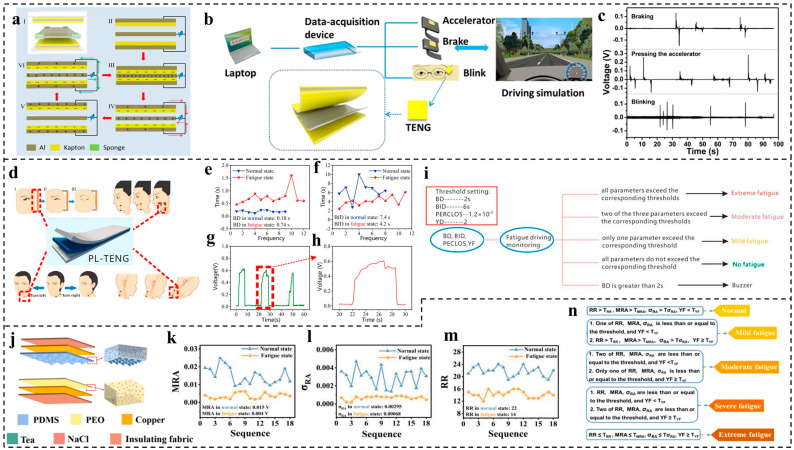
TENG-based sensor for fatigue detection. (**a**) The structure design of a TENG with an interlayer. (**b**) Schematic illustration of self-powered triboelectric sensors for driver behavior monitoring. (**c**) The recorded voltage signals of the braking process, pressing the accelerator, and blinking. Reproduced with permission [110]. Copyright 2018, Elsevier. (**d**) Schematic diagram of TENGs attached on eye corners, mouth to the lower jaw, back, and left of neck. (**e**) BD under normal driving and fatigue driving (within 1 min). (**f**) BID under normal driving and fatigue driving (within 1 min). (**g**) Yawning waveform within 1 min of fatigue driving. (**h**) A magnified yawning waveform. (**i**) Flow chart of fatigue driving judgment. Reproduced with permission [111]. Copyright 2020, Elsevier. (**j**) The internal structure of PP-TENGs. (**k**) The MRA, (**l**) σ_RA_, and (**m**) RR under normal and fatigue driving states. (**n**) The evaluation criteria of the driver fatigue degree [112]. Reproduced with permission. Copyright 2022, Elsevier.

**Figure 10 nanomaterials-14-00336-f010:**
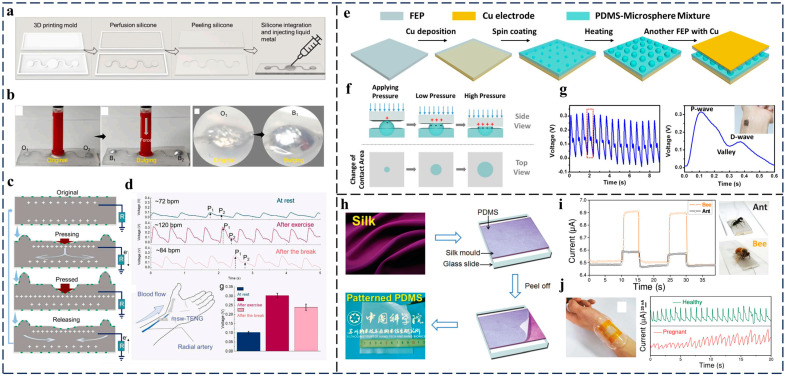
TENG-based sensor for wrist pulse monitoring. (**a**) The structure design of the msw-TENG. (**b**) Photograph of the msw-TENG after pressing in bulging state. (**c**) Working mechanism of the msw-TENG in pressed mode. (**d**) Output signal of the msw-TENG for monitoring radial artery pulse signals in different motion states. Reproduced with permission [46]. Copyright 2021, Elsevier. (**e**) Schematic diagram of the fabrication process and structure of the pressure sensor. (**f**) Deformation process of the pressure sensor when pressure is applied. (**g**) Voltage signals recorded during the pulse-rate-monitoring test. Reproduced with permission [52]. Copyright 2019, Elsevier. (**h**) Schematic of the fabrication process of flexible, patterned PDMS films. (**i**) Real-time electrical outputs of the e-skin constructed for the detection of a bee and an ant. (**j**) Pulse signals for monitoring wrist pulses of a healthy person and a pregnant woman. Reproduced with permission [53]. Copyright 2013, Wiley-VCH.

**Figure 11 nanomaterials-14-00336-f011:**
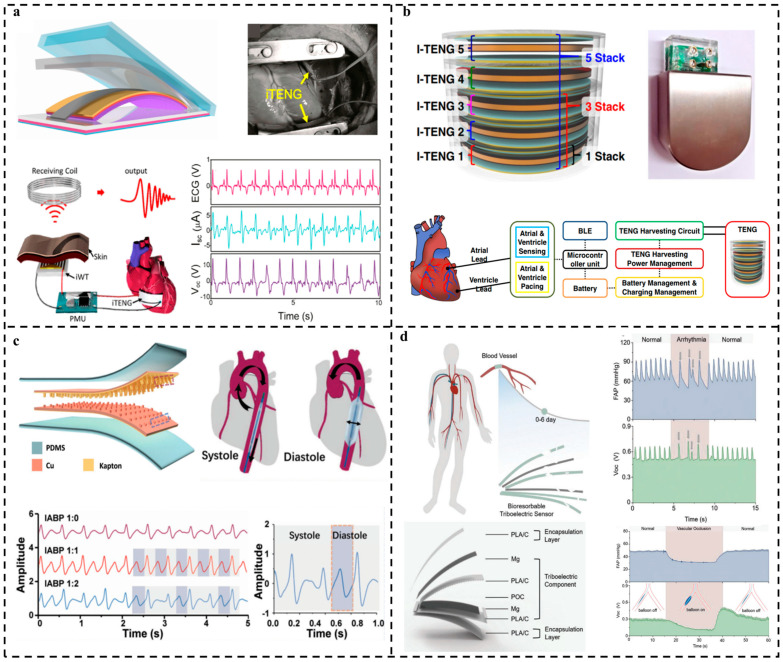
In vivo TENG-based sensor for preventive heart-health monitoring. (**a**) In vivo, self-powered, wireless cardiac monitoring via implantable TENG. Reproduced with permission [119]. Copyright 2016, American Chemical Society. (**b**) Self-rechargeable cardiac pacemaker system with TENG. Reproduced with permission [120]. Copyright 2021, Springer. (**c**) Self-powered pulse sensor for antidiastole of cardiovascular disease. Reproduced with permission [121]. Copyright 2021, Wiley-VCH. (**d**) A bioresorbable dynamic pressure sensor for cardiovascular postoperative care. Reproduced with permission [55]. Copyright 2021, Wiley-VCH.

**Table 1 nanomaterials-14-00336-t001:** Typical application functions of preventive health monitoring based on TENGs and their applied materials.

Year	Researchers	TENG Application	Used Materials	Function	Performance
2017	Jeon et al. [41]	Sensor arrays	PTFE and Kapton	Fall detection by evaluating the triggered number of TENG arrays	*V_oc_* of 162 V *I_sc_* of 22 µA 95.75% classification accuracy
2018	Lin et al. [66]	Smart insole	Rubber and copper	Fall detection by monitoring the signals generated by walking	Response time less than 56 ms
2021	Zhang et al. [44]	Smart insole	TPU-coated PES and conductive nickel fabric	Fall detection by monitoring the signals generated by walking	Sensing range more than 245 Kpa Durability more than 5000 cycles
2022	Kou et al. [51]	Smart pillow	PDMS and FEP	Fall detection on bed by monitoring the position of head	Sensitivity of 2.1 mV/Pa Durability more than 14,000 cycles
2022	Guo et al. [43]	Smart walking stick	Ecoflex, Nitrile, Al, and PTFE	Mobility disability evaluation motion status determination and fall detection enabled by AI	Average power density of 0.137 mW cm^−3^ at 0.083 Hz 100% classification accuracy
2022	Lu et al. [92]	Smart mask	FEP film and Al foil	Apnea alarming and switching on/off the operation of household appliances by monitoring respiration	Light weight of 4.7567 g
2023	Wang et al. [45]	Smart mask	Cotton fabric and polyethylene	Remote respiration monitoring by using advanced IoT technology	Maximum communication distance of 20 km
2015	Wen et al. [103]	Ethanol sensor	FEP and copper	Self-powered ethanol detection enabled by connecting a gas-sensing, rhombus-shaped Co_3_O_4_ nanorod array with TENG	Detection limit of 10 ppm Sensitivity around 0.15 ppm^−1^ 11 s of response time 20 s of recovery time
2019	Wang et al. [54]	Ammonia sensor	PDMS and Ce-doped ZnO-PANI film	Self-powered ammonia detection enabled by the chemical reaction between PANI and ammonia, which leads to the output change	Detection limit of 0.1 ppm Sensitivity of 1.1 ppm^−1^ 109 s of response time 233 s of recovery time
2020	Su et al. [100]	Acetone sensor	PTFE and nylon	Self-powered acetone detection by using chitosan and reduced graphene oxide (RGO) as sensitive materials on electrode, which can react with acetone	Detection limit lower than 2 ppm Sensitivity of 2.71 ppm^−1^
2018	Meng et al. [110]	Wearable sensor	Al foil and Kapton	Fatigue detection by monitoring the blinking and braking behavior of driver	*V_oc_* of 14 V *I_sc_* of 1.2 µA
2020	Lu et al. [111]	Wearable sensor	PDMS and human skin	Fatigue detection by attaching TENG sensors on different parts of driver for monitoring physiological signals	Durability more than 10,000 cycles *V_oc_* more than 200 V
2022	Wu et al. [46]	Pressure sensor	Silicone and EGaIn	Pulse information collection by attaching TENG on arterial pulse	Stretchability around 300% Durability more than 10,000 cycles
2021	Ouyang et al. [55]	Pressure sensor	PLA/C and Mg	Cardiovascular event identification by attaching TENG sensors on vascular wall	Durability more than 45,000 cycles 99% sterilization Sensitivity of 11 mV mmHg^−1^ Service life more than 5 days

## Data Availability

The data presented in this study are available on request from the corresponding author.

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
