# Peer review of "Triboelectric Nanogenerators for Preventive Health Monitoring"

_nanomaterials, 2024, doi:10.3390/nano14040336_

Round 1
Reviewer 1 Report
Comments and Suggestions for Authors
The paper provides a review of various ways in which triboelectric nanogenerators can be used as sensors within a healthcare setting. It provides a good and thorough overview of what has happened within this field. There is one thing that appears to be missing. The authors mention at several points in the paper that the advantage of TENGs is that they are self-powered so do not require external power sources. This is correct, what the authors do not discuss is how the signal is communicated to the outside world and where the power for this comes from. Does the TENG provide sufficient power to do this or is an external power source and communication device required before a TENG can effectively be used. This would negate the advantage that a TENG is self-powered.
Another major issue is the quality of the figures, they are really poor and impossible to understand.
Some minor comments:
- Line 33, ‘massive’ is the wrong word as it depends on the perspective with which you look.
- Line 50, ‘birth’ surely this must be conception, moreover, this contradicts a later statement that the triboelectric effect has been around for thousands of years.
Comments on the Quality of English LanguageThe quality of English is very poor and needs substantial work for the work to be acceptable.
Reviewer 2 Report
Comments and Suggestions for Authors
Journal Title: Nanomaterials
Manuscript Title: Triboelectric Nanogenerator for Preventive Health Monitoring
Manuscript ID: nanomaterials-2841193-peer-review-v1
Authors: Mang Gao et al.
The current work reviews the most recent progress in preventive health monitoring depending on triboelectric effects. The scientific content is rich, the exposition is logical and coherent, the main objection would be the fact that it does not necessarily fit into the specifics of the Nanomaterials Journal. However, it is up to the editors if this would be an impediment to the publication of this article in Nanomaterials. In my opinion the review might be published after minor revision according to the critical points expressed bellow:
-none of the 10 figures is legible, improved resolution images must be provided
-in order to better fit this journal, a subchapter describing nanomaterials used in the manufacture of triboelectric nanosensors, even if such reviews are presented in the literature
Reviewer 3 Report
Comments and Suggestions for Authors
The study describes a review on TriboElectric NanoGenerators (TENGs) for health monitoring. The device is a proof of concept for wearable electronics, IoT and health monitoring applications. The manuscript is well-organized and a valid survey of current devices is provided to readers. This study is suitable for expert community. Some Minor Revisions are suggested:
- The Figures are full including many sub-figures at low resolution. Please, revise this to consider supplementary section of figures.
- The nanodevices are well-described but nanomaterials used into devices are poorly described. Please, revise this aspect in order to suit manuscript to Nanomaterials journal.
Table 1 summarizes some typical applications of TENGs. However, the function of devices reported in literature is described and no technical indicator is listed in Table. Please, add a column reporting technical data of each described TENG.
After these Minor Revisions, the manuscript could be publishable.
Comments on the Quality of English LanguageMinor editing of English is suggested.
Round 2
Reviewer 1 Report
Comments and Suggestions for Authors
While I am happy with the majority of the corrections, the quality of the figures is still very poor.
Comments on the Quality of English LanguageThe English is fine.